# Self-DNA Inhibition in *Drosophila melanogaster* Development: Metabolomic Evidence of the Molecular Determinants

**DOI:** 10.3390/biology12111378

**Published:** 2023-10-27

**Authors:** Michele Colombo, Laura Grauso, Virginia Lanzotti, Guido Incerti, Adele Adamo, Aurora Storlazzi, Silvia Gigliotti, Stefano Mazzoleni

**Affiliations:** 1Institute of Biosciences and BioResources, National Research Council, Via Pietro Castellino 111, 80131 Napoli, Italy; michele.colombo@ibbr.cnr.it (M.C.); adele.adamo@ibbr.cnr.it (A.A.); aurora.storlazzi@ibbr.cnr.it (A.S.); 2Department of Agricultural Sciences, University of Naples Federico II, 80055 Portici, Italy; laura.grauso@unina.it (L.G.); virginia.lanzotti@unina.it (V.L.); 3Department of Agri-Food, Animal and Environmental Sciences (DI4A), University of Udine, 33100 Udine, Italy; guido.incerti@uniud.it

**Keywords:** self-DNA inhibition, model insect, bioactive metabolites, NMR analysis, LC-MS analysis, biocontrol

## Abstract

**Simple Summary:**

The article describes the inhibitory effects of self-DNA in the insect *Drosophila melanogaster*. The larvae fed with yeast containing their self-DNA showed a significant delay in their development and some increased mortality. The chemical analysis of these larvae showed changes of their metabolites’ composition, with the presence of some compounds known for their effects on insect development and/or fertility. In fact, the adult flies grown from self-DNA-treated larvae showed a strongly reduced egg deposition.

**Abstract:**

We investigated the effects of dietary delivered self-DNA in the model insect *Drosophila melanogaster*. Self-DNA administration resulted in low but significant lethality in Drosophila larvae and considerably extended the fly developmental time. This was characterized by the abnormal persistence of the larvae in the L2 and L3 stages, which largely accounted for the average 72 h delay observed in pupariation, as compared to controls. In addition, self-DNA exposure affected adult reproduction by markedly reducing both female fecundity and fertility, further demonstrating its impact on Drosophila developmental processes. The effects on the metabolites of *D. melanogaster* larvae after exposure to self-DNA were studied by NMR, LC-MS, and molecular networking. The results showed that self-DNA feeding reduces the amounts of all metabolites, particularly amino acids and N-acyl amino acids, which are known to act as lipid signal mediators. An increasing amount of phloroglucinol was found after self-DNA exposure and correlated to developmental delay and egg-laying suppression. Pidolate, a known intermediate in the γ-glutamyl cycle, also increased after exposure to self-DNA and correlated to the block of insect oogenesis.

## 1. Introduction

The existence of inhibitory effects of extracellular self-DNA on living organisms was first discovered in plants [1] and then extended to other living kingdoms [2,3], with demonstration in different model organisms such as *Arabidopsis thaliana* [4], *Caenorhabditis elegans* [5], and *Saccharomyces cerevisiae* [6].

In plants, the molecular mechanisms of the inhibitory process by self-DNA exposure have been investigated by transcriptomic [4,7], physiological [8,9,10], and metabolomic [11] approaches, showing a generalized significant reduction in metabolic activities with molecular responses typical of stress conditions, specifically including ROS production [9,10] and accumulation in the exposed cells of RNA constituents and precursors [11].

In animals, the only previous study on *C. elegans* highlighted the setting of larval developmental defects and increased mortality with magnification of the effects progressively increasing in subsequent generations [5]. In this model nematode, the observations of self-DNA impacts on egg fertility and fecundity clearly indicated an involvement of the germinal cell lines, whose mechanism is yet to be clarified.

This work extends the range of model organisms in the study of self-DNA inhibition to *Drosophila melanogaster* with the two main objectives of assessing the occurrence and relevance of the inhibitory effect and the investigation of the molecular determinants of the phenomenon by using high-throughput metabolomic analysis.

Figure 1 shows the schematic representation of the experimental workflow.

## 2. Materials and Methods

### 2.1. Yeast Library

#### 2.1.1. Generation of a *Drosophila melanogaster* Genomic Library in Yeast

The library was constructed by Bio S&T Inc. (6362 Trans Canada Route, Saint-Laurent, Quebec, Canada H4T 1X4). HMW genomic DNA extracted from third instar Drosophila larvae was partially digested with Sau3AI and cloned into the BamH1 site of the pGADT7 vector, which was initially transformed into *E. coli* DH10B host cells. The bacterial library was amplified by overnight culture at 37 °C on agar medium and recombinant plasmids were isolated and used to transform *S. cerevisiae* Y187 host cells. The obtained library in yeast (self-DNA library) was checked for quality (empty clone rate: 1/11; average insert size: 4.3 kb), distributed in aliquots, and shipped.

#### 2.1.2. Amplification of the Self-DNA Library

Yeast cells were plated on Petri dishes (150 × 10 mm) containing 100 mL SD agar medium (2% agar, 0.17% yeast nitrogen base, 0.5% ammonium sulfate, 2% glucose) supplemented with adenine, histidine, methionine, and tryptophan (40, 20, 20, 40 µg/mL, respectively), 2 × 10^5^ cells/dish, for a total of 25 dishes. Plates were incubated at 30 °C for 5 days to allow yeast colonies’ growth. Yeast cells were washed out in H_2_O, pelleted (5 min centrifugation at 3000 rpm, 12 °C, in 5810 R Eppendorf centrifuge equipped with A-4-62 swinging rotor), resuspended in H_2_O for absorbance reading, pelleted as above, and resuspended at a 40 OD_600_ concentration in SD medium supplemented with adenine, histidine, methionine, and tryptophan (same concentration as above) and 15% glycerol, to be finally dispensed in 1 mL aliquots (corresponding to about 6.5 × 10^8^ cells) and stored at −80 °C.

#### 2.1.3. Preparative Growth of Yeast for Administration to Flies

Single aliquots of the amplified self-DNA library were inoculated in 400 mL SD medium supplemented with adenine, histidine, methionine, and tryptophan (see above). Seeded flasks were incubated at 30 °C on a shaker set to 220 rpm for 48 h, up to a cell density of about 2 OD_600_. Yeast cells were pelleted by centrifugation at 5000 rpm for 20 min at 12 °C in a Beckman Avanti J-25 centrifuge equipped with a JA-14 rotor, resuspended in water, transferred into 15 mL Falcon tubes, re-pelleted (Eppendorf centrifuge 5810 R equipped with a A-4-62 swinging rotor, run at 3000 rpm for 10 min at 12 °C), stored O/N at −20 °C, and finally freeze-dried.

Control *S. cerevisiae* Y187 cells, containing the empty pGADT7 vector, were similarly prepared.

### 2.2. Fly Bioassays

#### 2.2.1. Drosophila Culture Conditions

Wild-type flies of the Oregon R strain were routinely maintained on a standard sucrose–cornmeal–yeast medium at 25 °C, in a 12 h light and 12 h dark cycle. The commercial yeast present in this medium was replaced by lab-made preparations of the self-DNA library or the control yeast for performing the bioassays. 

Preliminary tests were conducted to scale-down culture conditions without affecting larval growth and development. On the basis of these tests, bioassays were conducted with groups of 20 larvae housed in 11 × 100 mm glass tubes containing 1 mL culture medium. In addition, all the experiments were performed with a large excess of yeast, to ensure that both Drosophila larvae and adults could feed *ad libitum*.

#### 2.2.2. Toxicity Assays

To obtain synchronized embryos, 2-day-old flies were transferred to culture vials that fit 35 mm Petri dishes, containing agar–apple juice medium and a heap of commercial yeast to stimulate oviposition. Dishes were replaced twice a day for three days and embryos laid in a 4 h temporal window were collected on the fourth day. After thorough washing with water, they were transferred to fresh agar–apple juice dishes supplied with 7 mg (dry weight) of control yeast or a self-DNA library put on the surface and incubated at 25 °C until hatching.

Newborn first-instar larvae were picked from the plates and transferred to glass tubes containing 1 mL culture medium supplemented with an extra layer of yeast (5.4 mg dry weight of control yeast or the self-DNA library, as appropriate), in groups of 20 larvae/tube. Additional yeast aliquots were administered to developing larvae on the next four days, once per day.

Development was monitored by scoring the number of newly formed pupae and newly emerged adults at 12 h time intervals, starting from 300 larvae feeding on either control yeast or the self-DNA library. The duration of development from embryo to pupae was empirically determined as the interval (hours) between egg laying and the time when pupariation had reached 50% of the total number of pupae, both for the group fed control yeast and the group fed the self-DNA library (T1/2 control pupae and T1/2 self-DNA pupae, respectively). The larval developmental delay was then calculated as the difference between the duration of the larval stages for the self-DNA treatment and the duration of the larval stages for control treatment (T1/2 self-DNA pupae—T1/2 control pupae).

Lethality during larval stages was calculated as the ratio of total pupae to total L1 larvae.

#### 2.2.3. Time-Course of Larval Development

Newborn first-instar larvae were obtained from synchronized embryos and allowed to develop in culture tubes as described above. The progression through larval stages was scored at 24 h time intervals. At each time-point, larvae were collected from food, washed in cold PBS, transferred to microscope slides, and frozen at −20 °C for 10 min, until no movement was observed. Larvae pictures were taken with a Leica Microsystems GmbH (Wetzlar, Germany) MZ 9.5 microscope equipped with a Leica Microsystems GmbH IC80 HD digital camera controlled by the Leica Microsystems GmbH LAS EZ imaging software and were staged according to the morphology of the mouth hooks and of the anterior and posterior spiracles [12].

#### 2.2.4. Fecundity and Fertility Tests

Adult flies of 0–24 h deriving from larvae fed control yeast or the self-DNA library were collected from glass culture tubes and transferred to culture vials that fit 35 mm Petri dishes, containing agar–apple juice medium plus 20 mg of the respective control yeast or self-DNA library deposited on the surface. Dishes were replaced twice a day and, starting from the fourth day, were examined to count laid eggs and hatched larvae. Fecundity, scored as the average number of eggs laid per day by a single female, was assessed by recording, for four consecutive days, the number of eggs laid by a group of 25 females fed control yeast and a group of 21 females fed the self-DNA library. Fertility, scored as the rate of embryo survival up to hatch, was assessed by recording the total number of L1 larvae detected on all Petri dishes used for egg count.

### 2.3. Metabolomic Analysis

#### 2.3.1. Metabolomic Extraction

*D. melanogaster* larvae were extracted with dichloromethane (1 mL), sonicated for 10 min, and centrifuged at 7000 rpm for 10 min at room temperature. The dichloromethane extracts (apolar extracts) thus obtained were taken to dryness and stored at −80 °C until analysis. The larvae were then extracted with a 1:1 solution of methanol: water, sonicated for 10 min, and centrifuged at 7000 rpm for 10 min at room temperature. The polar extract thus obtained was divided into two aliquots, one for LC-MS analysis and the other one for NMR analysis. The two aliquots were dried under vacuum and stored at −80 °C until analysis.

#### 2.3.2. NMR Analysis

Dried polar extracts of *D. melanogaster* were run in D_2_O (99.9%) adding 0.2 mg/mL DSS as an internal standard. Dried apolar extracts were run in CDCl_3_ adding 0.2 mg/mL TMS as an internal standard. NMR spectra were registered on a Bruker Avance NEO 600 MHz spectrometer by using a z-gradient 5 mm ^1^H/^13^C/_15_N/^31^P cryogenic probe head. Chemical shifts were referred to DSS signal (Δ 0.00 ppm). The spectra were processed using the iNMR program (https://www.inmr.net (accessed on 6 March 2023)). In total, 12 spectra (4 animal populations × 3 replicates) were acquired. Quantification was performed by integration of the signals relative to the internal standard, DSS. The residual H_2_O signal was suppressed by presaturation [11].

#### 2.3.3. LC-MS Analysis

Polar extracts of *D. melanogaster* were dissolved in 500 μL of a mixture of methanol: water 1:1, then were injected to LC-MS. LC-MS and LC-MS/MS experiments were run on a Thermo LTQ Orbitrap XL mass spectrometer (Thermo Fisher Scientific Spa, Rodano, Italy), and a Thermo U3000 HPLC. A Kinetex 5 µm, 50 × 2.1 mm C18 column (Phenomenex, Torrance, CA, USA) was used with 0.1% formic acid in H_2_O as solvent A and CH_3_OH as solvent B. The gradient elution was performed as follows: 5% B 1 min, 5% to 100% B over 40 min, hold 10 min, flow rate 0.2 mL/min. The injection volume was set to 5 μL.

High-resolution mass spectra (HR-MS) and high-resolution MS/MS spectra (HR-MS/MS) were run in positive ion mode in the range of *m/z* 100–2000 with resolution set to 60,000 as previously reported [13].

#### 2.3.4. Molecular Network Analysis

The data of the LC-MS/MS were processed using mzMine version 2.53. The molecular network was obtained by the Feature-Based Molecular Networking (FBMN) workflow [14] on GNPS [15]. Data were filtered by removing the MS/MS peaks in the ±17 Da interval of the *m/z* precursor. MS/MS spectra were filtered by choosing the top 6 fragments in the ±50 Da interval. The precursor ion mass tolerance was set to 0.02 Da, while the MS/MS fragment ion tolerance was set to 0.02 Da. A molecular network was then obtained as previously reported [11].

The molecular network was visualized by Cytoscape software 3.9.1 [16]. Nodes are represented as pie charts. Red slices indicate the sum of peak integrations of all self-DNA treatment samples, light blue slices indicate the sum of peak integrations of all other analyzed samples.

### 2.4. Data Analysis

Data on inhibition experiments were tested for significant differences between self-DNA treatment and the control considering larval lethality (% of all larvae), pupal lethality (% of all pupae), fertility (number of eggs per mother per day), and hatching rate (% of all eggs) as dependent variables. For each dependent variable, a *t*-test for independent unpaired samples was run at alpha = 0.05 as the threshold value for statistical significance, after testing for the parametric assumptions of normality and homoscedasticity, with Shapiro–Wilk’s and Levene’s tests, respectively.

## 3. Results

### 3.1. Inhibitory Effects of Self-DNA on Larval Growth

We previously reported on the successful usage of a recombinant microbial library for the administration of self-DNA to the nematode *Caenorhabditis elegans*. Therefore, we decided to employ an analogous strategy in *Drosophila melanogaster* and selected *Saccharomyces cerevisiae* as a suitable vector for the delivery of self-DNA to flies. This choice was based on the fact that natural Drosophila habitats share the common characteristic of an abundant yeast microflora, which produces volatiles that are highly attractive to flies: among them, *Saccharomyces cerevisiae* is the most attractive to *D. melanogaster* and is routinely used as a major component of the laboratory fly diet [17,18,19]. In addition, it was previously reported that genetically modified *S. cerevisiae* can act as a vehicle of dsRNA molecules and induce RNAi in *D. suzukii* upon ingestion [20].

A *D. melanogaster* genomic library was generated in the multicopy, 2 micron pGADT7 plasmid vector and transformed into the *S. cerevisiae* Y187 strain. Recombinant yeast cells were then grown in large liquid cultures to produce the biomass needed for replacing commercial yeast in the preparation of the standard Drosophila diet. This diet was used to feed Drosophila wild-type flies and test the effects of self-DNA exposure. A control diet containing *S. cerevisiae* Y187 cells with an empty pGADT7 plasmid vector was prepared and used in parallel.

We first tested the impact of the self-DNA diet on fly survival during development. To this aim, we assessed pupae formation and adult eclosion rates. When fed the self-DNA library, 8.7% of larvae died before reaching the pupal stage, thus showing a mortality rate that was significantly different from the 4.7% baseline observed in the control group (unpaired *t*-test: *p* < 0.05) (Figure 2A). Thus, self-DNA exposure had a clear detrimental effect on Drosophila development. However, this effect was restricted to the feeding stages, since pupae deriving from larvae fed the self-DNA or the control diet displayed comparable survival to adulthood (Figure 2B).

When performing the survival experiment described above, we also monitored the time needed for the transition from larvae to pupae and from pupae to adults, by scoring the number of newly formed pupae and newly emerged adults at 12 h time intervals. Self-DNA library administration induced a considerable delay in larval development, whose duration extended by 72 h, as compared to the control treatment (Figure 2C). In contrast, the duration of the pupal stage for the self-DNA-treated group was similar to that of the control, further indicating that self-DNA ingestion caused inhibitory effects on larvae but did not affect metamorphosis. Thus, the overall delay that was observed in adult emergence upon self-DNA treatment simply reflected the pupariation delay (Figure 2D).

To precisely describe this developmental delay, we performed a time-course experiment in which we scored the progression through larval stages at 24 h time intervals (Figure 3A). At 24 and 48 h after egg-laying (AEL), self-DNA fed larvae were comparable to control larvae, both in size and developmental stage (Figure 3B). At 72 h AEL, however, while 80% of control larvae were in the late L2 stage and the remaining 20% had molted to the L3 stage, self-DNA fed larvae displayed a reduced size compared to their time-matched controls, as they were mostly at the early L2 stage and a small fraction was still in the L1 stage, indicating that early larval development slows down in the presence of self-DNA (Figure 3B). This inhibitory effect persisted at 96 h AEL, when 100% of control larvae were in the L3 stage, but all self-DNA fed larvae were in the L2 stage (Figure 3B). They started to molt only during the following 24 h: at 120 h AEL, while control larvae had fully developed, 79% of self-DNA fed larvae were at the early L3 stage (Figure 3B). Observations made at later time-points, 144 h and 168 h AEL, documented the transition to the wandering stage and final pupariation in control larvae, while self-DNA-fed larvae were still proceeding in their development through the L3 stage (Figure 3B).

Taken together, these data indicate that self-DNA oral administration interferes with the time-regulated progression through larval developmental stages, mainly causing prolonged permanence in the L2 and L3 stages, and also inducing a low but significant rate of mortality.

### 3.2. NMR Metabolite Profiling after Self-DNA Exposure

Changes in the metabolite profile of *D. melanogaster* larvae after exposure to self-DNA were preliminarily evaluated by NMR metabolomics. The larvae were fed with either a self-DNA library or control yeast and sampled at 72 h and 96 h AEL (self-DNA: samples S1a, S2a, S3a/S1b, S2b, S3b, respectively; control: C1a, C2a, C3a/C1b, C2b, C3b, respectively) (see Materials and Methods). Extraction with dichloromethane and methanol/water 1:1 was performed to obtain apolar and polar extracts, respectively, which were analyzed by NMR and LC-MS combined with molecular networking.

The ^1^H NMR spectra of apolar extracts showed very similar metabolite profiles characterized by fatty acids as major compounds in the analyzed samples (Appendix A). Peak integration confirmed the similarity of the chemical composition, thus indicating that self-DNA exposure did not induce significant qualitative or quantitative changes in apolar metabolites of *D. melanogaster* larvae.

In contrast, the preliminary ^1^H NMR analyses of polar extracts evidenced a more complex chemical composition, characterized by the presence of compounds belonging to different chemical classes and by clear differences in the metabolite profiles between the treatments (Figure 4). Therefore, we focused our study on polar extracts.

The metabolomic fingerprinting analysis showed an evident decrease in metabolite production in self-DNA treated samples (samples Sa and Sb, triplicates 1–3) when compared to controls (samples Ca and Cb, triplicates 1–3). In fact, all signals decreased in their intensity and the decrease was directly proportional to the duration of the treatment, so that the minimum signal intensity was reached at a longer exposure time (samples Sb, triplicates 1–3) (Figure 4). In particular, a decrease was observed in the signals ascribable to carbohydrates (δ 3.20 and 5.40 interval). In addition, the signals of aliphatic and aromatic compounds (δ 0.80–3.00 and δ 6.50–8.00, respectively) were decreasing in the self-DNA treatments.

### 3.3. LC-MS and Molecular Network

To evaluate the changes in specific metabolites in the extracts from a qualitative and quantitative point of view, LC-MS of the treated larvae were acquired (Appendix A). Inspection of the LC-MS datasets was aided by the application of the molecular networking approach [15]. Appendix A shows a whole picture of the obtained data, including peak clusters and single metabolite peaks. Among these data, three groups of peaks, indicated as I, II, and III (Figure 5) were selected in which nodes represent the compounds detected in the insect extracts. The red color indicates compounds detected in the self-DNA treatment (S), where the light red corresponds to samples collected at 72 h AEL (samples Sa, replicates 1–3), while the red color shows the compounds detected at 96 h AEL (samples Sb, replicates 1–3). The light blue color was used to mark compounds obtained by control treatments collected at 72 h AEL (samples Ca, replicates 1–3) and the blue color the same compounds at 96 h AEL (samples Cb, replicates 1–3). The size of the node, indicating the detected ion intensity in the positive ionization mode, directly relates to the amount of each specific compound in the extracts. Figure 5 shows groups I and II, including the metabolites produced at higher amounts in control samples, and group III, selecting metabolites specifically accumulated in insect samples upon self-DNA treatment.

A peak-by-peak analysis of cluster I (Figure 5 and Table 1) pointed to the identification of N acyl- amino acids. Thus, the amino acid histidine was found connected via an amide bond to lauric (C 12:0, *m/z* 338.2437 [M + H]^+^, C_18_H_32_O_3_N_3_, −0.35 ppm), myristic (C 14:0, *m/z* 366.2753 [M + H]^+^, C_20_H_36_O_3_N_3_, −1.00 ppm), myristoleic (C 14:1, *m/z* 364.2593 [M + H]^+^, C_20_H_34_O_3_N_3_, −0.46 ppm), and palmitoleic (C 16:1, *m/z* 392.2908 [M + H]^+^, C_22_H_38_O_3_N_3_, 0.08 ppm) acids. The amino acid tyrosine was detected as myristoyl (*m/z* 392.2797 [M + H]^+^, C_23_H_38_O_4_N, −0.97 ppm) and myristoleyl (*m/z* 390.2637 [M + H]^+^, C_23_H_36_O_4_N, −0.47 ppm) derivatives. Valine and phenylalanine were found as myristoyl derivatives (*m/z* 328.2846 [M + H]^+^, C_19_H_38_O_3_N, −0.14 ppm and *m/z* 376.2845 [M + H]^+^, C_23_H_38_O_3_N, −0.42 ppm, respectively). Aspartic acid was connected to myristic (*m/z* 344.2420 [M + H]^+^, C_18_H_34_O_5_N, −3.45 ppm) and myristoleic (*m/z* 342.2274 [M + H]^+^, C_18_H_32_O_5_N, −0.20 ppm) acids. Finally, glutamic acid was found linked to myristic (*m/z* 358.2588 [M + H]^+^, C_19_H_36_O_5_N, −0.02 ppm), myristoleic (*m/z* 356.2430 [M + H]^+^, C_19_H_34_O_5_N, −0.42 ppm), palmitic (C 16:0, *m/z* 386.2905 [M + H]^+^, C_21_H_40_O_5_N, 0.77 ppm), palmitoleic (*m/z* 384.2743 [M + H]^+^, C_21_H_38_O_5_N, −0.39 ppm), and oleic (C 18:1, *m/z* 412.3055 [M + H]^+^, C_23_H_42_O_5_N, −0.60 ppm) acids.

Group II shown in Figure 5 and Table 1 reports compounds belonging to the class of amino acids. In addition to the six amino acids already found as N-acyl derivatives and here detected as free molecules, the following nine amino acids were identified: proline, threonine, isoleucine, leucine, asparagine, lysine, methionine, arginine, and tryptophan (see Table 1).

Group III contained the compounds whose quantity increased in the self-DNA treatments (Figure 5 and Table 1). The first compound, increasing significantly in self-DNA samples, showed a molecular formula of C_6_H_7_O_3_ (−1.34 ppm), *m/z* 127.0388 [M + H]^+^. The compound was unambiguously identified as phloroglucinol on the basis of its fragmentation peaks observed in the MS/MS spectrum. A second compound increasing in self-DNA-treated larvae was identified as pidolate based on the *m/z* value at 130.0497 [M + H]^+^, C_5_H_8_O_3_N (−1.30 ppm) and fragmentation peaks in the MS/MS spectrum. The node at *m/z* 147.0763 [M + H]^+^ corresponded to the molecular formula of C_5_H_11_O_3_N_2_ (−0.80 ppm) and was indicative of glutamine. The node at *m/z* 268.1040 [M + H]^+^, C_10_H_14_O_4_N_5_ (−0.11 ppm), was assigned to the nucleoside adenosine. The node at *m/z* 203.0525, present as a sodium adduct [M + Na]^+^, corresponded to the molecular formula of C_6_H_12_O_6_Na and was indicative of the sugar glucose (−0.19 ppm). The node at *m/z* 357.2747 [M + H]^+^, C_19_H_37_O_4_N_2_ (−0.23 ppm) indicated the compound 5-amino-5-oxo-2(tetradecanoylamino) pentanoic acid, while the node at *m/z* 438.3787, present as the ammonium ion [M + NH_4_]^+^, C_21_H_50_O_5_N_4_ (2.57 ppm), was due to an acyl amino acid. Finally, the node at *m/z* 426.2613 [M + H]^+^ corresponded to the molecular formula of C_19_H_41_O_7_NP (−0.50 ppm) and was identified as lysophosphatidylethanolamine (C14:0). The chemical structure of the identified compounds is reported in Figure 6.

### 3.4. Inhibitory Effects of Self-DNA on Adult Reproduction

The finding that self-DNA ingestion affected larval development and metabolite profile prompted us to test if it could also impair fly reproduction. To this aim, we scored the number of eggs laid by sexually-mature females flies that had been continuously exposed to self-DNA through larval development, post-eclosion adult life, and mating, as well as the number of larvae that hatched from these eggs. The obtained results indicated that self-DNA treatment strongly reduced the reproductive potential of Drosophila (Figure 7A,B). In fact, control flies produced on average 4.87 eggs per day, while flies fed the self-DNA diet laid only 1.36 eggs (unpaired *t*-test: *p* < 0.0001) (Figure 7C). Moreover, the hatching rate decreased on average from 46.13% to 4.29% (unpaired *t*-test: *p* < 0.0001) (Figure 7D). Therefore, the inhibitory action exerted by self-DNA affected both Drosophila fecundity and fertility.

## 4. Discussion

### 4.1. Biological Assays Highlight Target Processes of Self-DNA Inhibition

Here, we reported on the developmental and reproductive traits of the model insect *Drosophila melanogaster* as affected by the oral administration of self-DNA and described the impact of self-DNA exposure on the metabolic profiles of Drosophila larval stages.

In our experimental setting, flies were fed recombinant yeast cells vectoring Drosophila DNA fragments. This treatment triggered distinct biological responses, thus confirming that the usage of genomic libraries generated in suitable microbial hosts is an effective strategy to deliver self-DNA molecules to target organisms [5].

Self-DNA administration had adverse effects on Drosophila developmental progression and strongly reduced the fly offspring. Similar effects were previously reported in different organisms, either uni- or multi-cellular, which responded to conspecific fragmented genomic DNA present in the culture medium by reducing their growth and performance [1,2,3,4,5,7,8,9,10]. Therefore, the inhibitory phenotypes we described in Drosophila further support the notion that environmental DNA molecules play evolutionarily conserved regulatory roles that have been largely overlooked thus far [2].

The impact of self-DNA exposure on Drosophila development mainly manifested in a marked delay of the onset of metamorphosis. However, survival was also affected, as indicated by the small but significant fraction of larvae that died in the presence of self-DNA in the diet.

Developmental delay and developmental stalling or mortality are typical phenotypes resulting from Drosophila exposure to toxic agents and other types of environmental stress [21,22]. In our self-DNA bioassays, these phenotypes were restricted to larval stages, as pupae completed their development at a rate and in a time-frame comparable to controls. Although we cannot exclude the possibility that self-DNA affects development in a stage-specific manner, we believe that self-DNA molecules introduced via larval feeding of recombinant yeast cells did not persist in the fly through metamorphosis. In fact, biologically active self-DNA molecules might undergo fast decay, as we also observed that the induction of larval developmental delay required a daily supply of live recombinant yeast cells.

The developmental delay highlighted in larvae exposed to self-DNA was associated with a reduced growth rate. In *Drosophila melanogaster*, developmental progression and systemic growth are regulated by genetic programs that coordinate these processes with environmental conditions [23,24]. Central components of these adaptation mechanisms are the ecdysone and insulin pathways.

Insulin signaling, the crucial regulator of tissue growth, is activated when sensing of nutrients, in particular of amino acids, in the fat body and the glia of the blood–brain barrier (BBB) promotes the expression and release of Drosophila insulin-like peptides (DILPs) by insulin-producing cells (IPCs) located in the brain [25,26,27]. Therefore, the finding that slowly growing larvae feeding on the diet containing self-DNA molecules had reduced amino acid levels, as detected by our metabolomic analyses, likely indicates that insulin signaling was impaired in these larvae. This is also supported by the observation that glucose levels exhibited an opposite variation and increased in response to self-DNA exposure. In fact, it has been demonstrated that the manipulation of insulin signaling, either by the ablation of IPCs in the Drosophila brain or by the deletion of a group of DILP genes, resulted in a hyperglycemic phenotype [28,29]. Although our extracts derived from whole larvae, it can be assumed that free glucose measurements in these samples closely reflected circulating glucose levels [30].

Amino acid levels might decrease upon self-DNA exposure because of reduced feeding. Developing larvae might in fact undergo this behavioral adaptation as a protective strategy to cope with self-DNA toxicity. Alternatively, the inhibitory effects of self-DNA might have a specific impact on food processing. Future analyses will allow us to distinguish between these two alternatives. However, it is interesting to note that the metabolomic profiling of self-DNA-treated larvae identified a potential effector of the growth-inhibitory action of self-DNA. Actually, larvae fed self-DNA displayed a significant increase in phloroglucinol, a polyphenol compound, as compared with controls. This finding correlates with previously reported data demonstrating that phloroglucinol affected larval growth in the insect *Bactrocera cucurbitae* (Coquillett) (Diptera: Tephritidae) when incorporated in the artificial diet, thus exhibiting an antibiosis effect [31]. Moreover, phloroglucinol was shown to inhibit cell growth by suppressing insulin signaling in human colon cancer HT-29 cells [32].

Growth control by the insulin pathway is tightly associated with developmental progression, which requires elevated pulses of the steroid hormone ecdysone prior to each molt and at the larval to pupal transition to specifically set the duration of the growth period. Therefore, crosstalk between insulin and ecdysone signaling allows proper coordination of growth and developmental timing [26,33,34,35,36]. Notably, this interplay also ensures the maintenance of body proportions when the growth of one body organ is impaired. It has been shown that injury of the imaginal discs (the adult tissue primordia that are present in the soma of the larva) due to irradiation, mechanical damage, localized induction of cell death, or the activation of apoptosis induces a delay in pupariation [37,38,39]. The signal produced by damaged imaginal tissues is an insulin-like peptide (DILP8) that inhibits ecdysone production through a central brain relay, to allow regeneration and compensatory growth [40,41,42,43,44]. In addition, during the extended larval period, intact tissues stop growing until the injured or delayed tissue catches up in development [45].

Strikingly, DNA damage and the activation of p53/CEP-1-dependent apoptosis have been shown to occur in gonadal germ cells of self-DNA-treated *C. elegans* worms [5]. Therefore, it will be interesting to assess whether self-DNA exposure might induce similar effects in Drosophila imaginal tissues at the larval stage, thus triggering systemic regulation of both the developmental time and growth rate by the insulin and ecdysone pathways.

Self-DNA exposure also inhibited Drosophila reproduction, as both the number of eggs laid by adult females and the number of larvae hatching from these eggs were strongly reduced when flies were reared in the presence of the recombinant yeast. These results suggest that self-DNA affects the Drosophila developmental processes that underlie the production of functional gametes.

### 4.2. Metabolomics Clearly Indicates Molecular Agents of Self-DNA Inhibition

Potential mechanisms leading to oogenesis failure upon self-DNA oral administration can be envisaged on the basis of the obtained metabolomic data.

Along with glutamine, the related pyroglutamate, commonly known as pidolate, was also shown to be present at higher amounts in the self-DNA samples. Pidolate is a ubiquitous but little-studied natural amino acid derivative. It is a known intermediate in the γ-glutamyl cycle, a metabolic pathway responsible for the biosynthesis and degradation of glutathione, and is thus associated with redox imbalance [46].

Central components of the γ-glutamyl cycle are γ-glutamyl transpeptidases, highly conserved proteins that catalyze the first step of glutathione degradation [47]. Most of the knowledge relating to them comes from studies conducted in mammals, which indicated the significant impact of such proteins on the delicate balance existing between the mechanisms promoting cell survival and those inducing cell death [48]. Although very little information is available on the physiological role of γ-glutamyl transpeptidases in insects, functional studies of these proteins in the parasitoid *Aphidius ervi* demonstrated their ability to affect oogenesis by provoking apoptosis in the ovaries of the host aphid [49,50].

Thus, the accumulation of pidolate in *D. melanogaster* after the oral administration of self-DNA, along with previous studies on the γ-glutamyl cycle, suggests a potential role of this metabolite in the observed decline in female fertility.

Phloroglucinol increase, as detected in the extracts of Drosophila fed with self-DNA, might also induce oogenesis defects.

Phloroglucinol has been shown to inhibit eggshell peroxidase (ESP), the enzyme responsible of the eggshell hardening process. In fact, it was demonstrated that the addition of this metabolite to the diet of *D. melanogaster* adults led to the deposition of both empty shells and chorion-less eggs [51]. In addition, a very recent study showed that phloroglucinol completely blocked oviposition and increased stress tolerance [52].

Therefore, phloroglucinol might be generated in response to stress signals induced by self-DNA exposure and mediate the reproductive arrest associated with this treatment.

As mentioned above, several pieces of data indicate that phloroglucinol interferes with the insulin signaling pathway in human cancer cells [34]. It is not known if this compound can similarly act in insects, but if this was true, it might have wide impact on Drosophila oogenesis. In fact, it was demonstrated that insulin levels control female germline stem cell proliferation and maintenance and trigger vitellogenesis from the fat body in response to the nutritional status [53,54,55,56].

Our current experimental setting did not allow us to obtain a detailed morphological description of the oogenesis defects induced by self-DNA exposure, which was hindered by the occurrence of a number of alterations also in the ovaries of control flies, likely due to the suboptimal nutritional value of the used yeast strain. Future work will allow to circumvent this problem and clear the whole picture. However, the finding that the number of laid eggs decreased in time only in self-DNA-treated flies suggests that self-DNA inhibition might affect germline stem cells, which are necessary to ensure a continuous supply of differentiated cells to sustain fertility.

To the best of our knowledge, this is the first report on metabolite profiling in an animal species exposed to extracellular self-DNA. The NMR data presented here demonstrated that the treatment with self-DNA specifically affects Drosophila metabolism by reducing the production of all classes of cellular metabolites. This evidence was very clear both by the reduced yield of organic extracts after self-DNA treatment and by the analysis of the NMR spectra, whose signals showed a significant decrease in intensity in self-DNA treated samples as compared to control treatments. Moreover, the decline in metabolite yields was proportional to self-DNA exposure time.

The reduction of metabolite production observed in *Drosophila melanogaster* upon self-DNA administration agrees with previous findings obtained by transcriptomic [4] and metabolomic [11] studies in *A. thaliana,* where exposure to self-DNA resulted in a dramatic decrease in gene expression and a consequent drop in plant metabolites.

The comparison of LC-MS data of control and self-DNA-treated samples clearly evidenced in the latter a significant decrease in the content of amino acids and their N-acyl derivatives. This was shown by the molecular networking data where cluster I and group II included metabolites present at lower amounts in self-DNA samples.

Fatty acids amides are well-known metabolites in *D. melanogaster* and have been shown to be a family of potent cell signaling lipids [57]. Specifically, it was demonstrated that N-acylation of biogenic amines is a critical process for neurotransmitter inactivation, cuticle sclerotization, and melatonin biosynthesis in *D. melanogaster* [58,59,60]. Thus, the significant decrease in these metabolites in self-DNA-treated samples may indicate the partial inactivation of these metabolic processes.

While negatively affecting overall metabolite production in Drosophila larvae, self-DNA exposure also resulted in selective accumulation of specific metabolites, listed in group III. This group includes compounds based on a different skeleton: ubiquitous primary metabolites such as glucose, adenosine, and glutamine, the already-mentioned phloroglucinol and pidolate, a substituted carboxylic acid derivative, 5-amino-5-oxo-2 (tetradecanoylamino) pentanoic acid, and a fatty acid derivative, lysophosphatidylethanolamine (LPE) (C14:0). Interestingly, the latter compound can serve as a precursor of phosphatidylethanolamine (PE) and the balance between these two phospholipid species has been implicated in insulin sensitivity and regulation of feeding behavior in Drosophila [61]. In addition, increased levels of adenosine have been functionally linked to the suppression of insulin signaling in response to external stressors in the fly [62]. 

In summary, metabolomic analyses of self-DNA-treated flies delineated a clear picture of metabolic disturbance, which nicely correlated with the observed developmental delay and also matched with the negative impact exerted by self-DNA on reproduction. However, the toxicity effects of the reported specific metabolites are not enough to explain the observed major impact of self-DNA on Drosophila developmental processes. It is reasonable to think that DNA synthesis is also blocked or greatly slowed down during the buildup of DNA metabolites in the Drosophila body, and we may likely assume that DNA polymerase activity is reduced and therefore both larval development and adult fertility are inhibited in the case of a DNA-rich diet. Such an assumption is consistent with our latest findings in the different model organism *S. cerevisiae* [6], in which we reported a growth cycle arrest in the S phase when cells are exposed to self-DNA in their growth medium.

## 5. Conclusions

In this study, we clearly showed that inhibition by self-DNA occurs in *Drosophila melanogaster*. In particular, feeding on self-DNA induces larval lethality, delays larval development, and reduces egg deposition and fertility. Such results, given the importance of the model organism targeted by our experimental work, are of particular relevance, as they extend over previous findings on different biological models in fungi, plants and animals, including *Saccharomyces cerevisiae*, *Arabidopsis thaliana*, and *Caenorhabditis elegans*. In general, our results are intriguing considering further fields of potential applications, such as testing in model mammals and even human research, to control ecto- and endo-parasitosis by directing self-DNA extracted from parasite species to conspecific targets. Moreover, ongoing research of major interest regards investigations of the effects of DNA from cancer tissues as inhibitors of cell proliferation.

Considering the molecular mechanisms underlying the inhibitory effects of self-DNA, high-throughput metabolomic analysis by ^1^H NMR and LC-MS highlighted a generalized reduction of all classes of cellular metabolites, and, by contrast, a small number of molecular compounds whose cellular content increased under exposure to self-DNA. Among these, pyroglutamate, commonly known as pidolate, and phloroglucinol are mechanistically involved in the inhibitory process of both oogenesis and oviposition, with the latter compound also being known for its antibiosis activity on dipteran larvae. As such, this is the first study clearly indicating molecular causal agents of self-DNA inhibition.

## 6. Patents

WO2014020624—Composition comprising nucleic acids of parasitic, pathogenic, or weed biological systems for inhibiting and/or controlling the growth of said systems.

IT202100021392A1—Improved inhibitory DNA compositions and use thereof, in particular integrated with metabolic treatment to enhance inhibitory effects.

## Figures and Tables

**Figure 1 biology-12-01378-f001:**
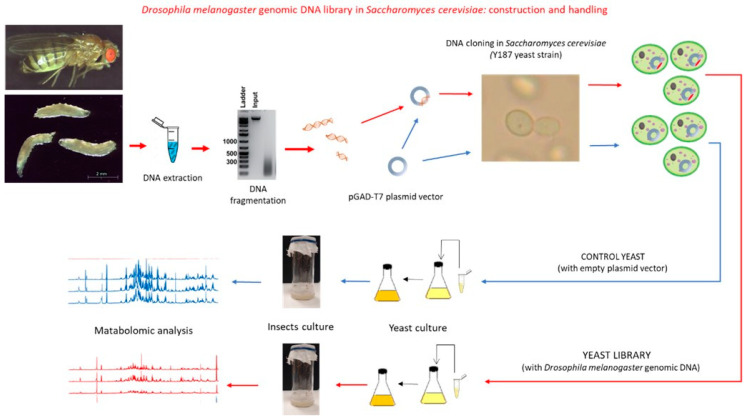
Schematic representation of the experimental set-up.

**Figure 2 biology-12-01378-f002:**
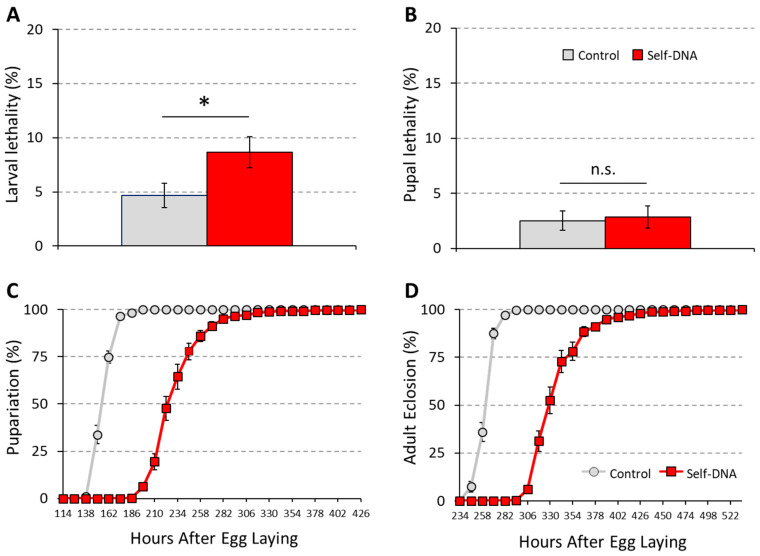
Increase in larval lethality (**A**), *: *t*-test for independent samples, *p* < 0.05, absence of significant effect on pupal viability (**B**), n.s.: *t*-test for independent samples, *p* > 0.05 and delay of larval development (**C**,**D**) in *Drosophila melanogaster* by self-DNA feeding.

**Figure 3 biology-12-01378-f003:**
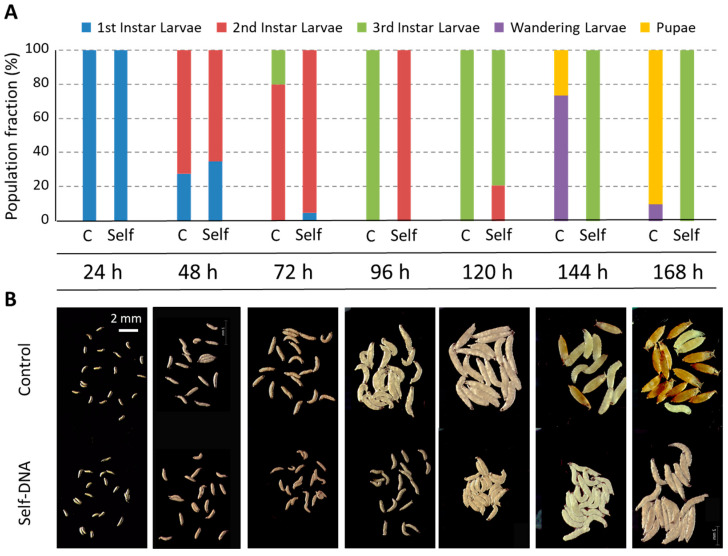
(**A**) Detail of larval developmental delay in *Drosophila melanogaster* by self-DNA feeding (Self), as compared to the control (C) along 7 observation times, which are expressed as hours AEL. (**B**) Representative illustrative pictures (scale bar in top left panel).

**Figure 4 biology-12-01378-f004:**
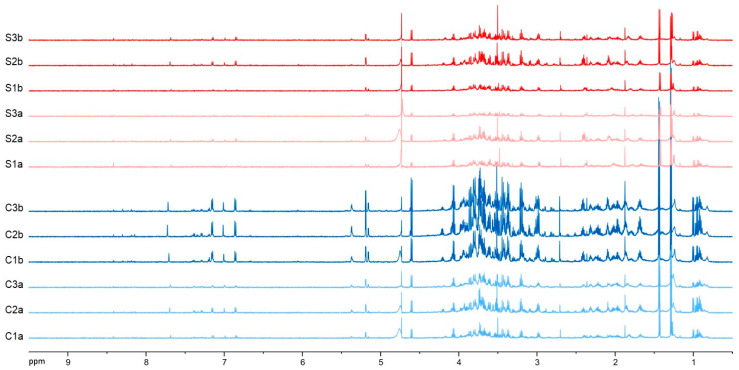
^1^H NMR spectra of the different *D. melanogaster* larval extracts. In red self-DNA samples obtained at longer exposure time (S1b, S2b, and S3b); in light red self-DNA samples at shorter exposure time (S1a, S2a, and S3a). In blue control samples at longer exposure time (C1b, C2b, and C3b); in light blue control samples at shorter exposure time (C1a, C2a, and C3a). Spectra were run at 600 MHz in D_2_O.

**Figure 5 biology-12-01378-f005:**
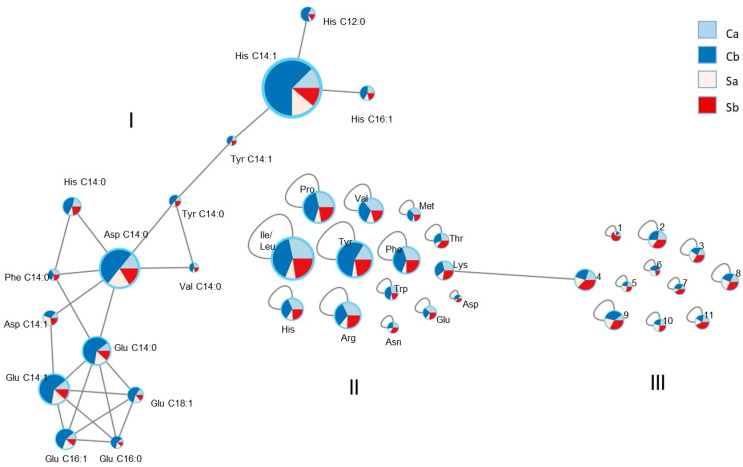
Molecular network based on the HRESI-LC-MS analysis of polar extracts. Nodes are reported as pie charts: light red and red slices indicate the sum of peak integrations of self-DNA treatment samples (Sa and Sb, respectively), light blue and blue slices indicate the sum of peak integrations of control samples (Ca and Cb, respectively). Node size directly relates to the peak integration and consequently to the amount of the specific metabolite. Group I is the cluster of compounds with similar structures prevailing in control conditions. Group II refers to different metabolites specifically accumulated in control samples. Group III refers to metabolites specifically accumulated in self-DNA-treated samples. Full compound names, referred in this figure by either abbreviations or numbers, are listed in Table 1.

**Figure 6 biology-12-01378-f006:**
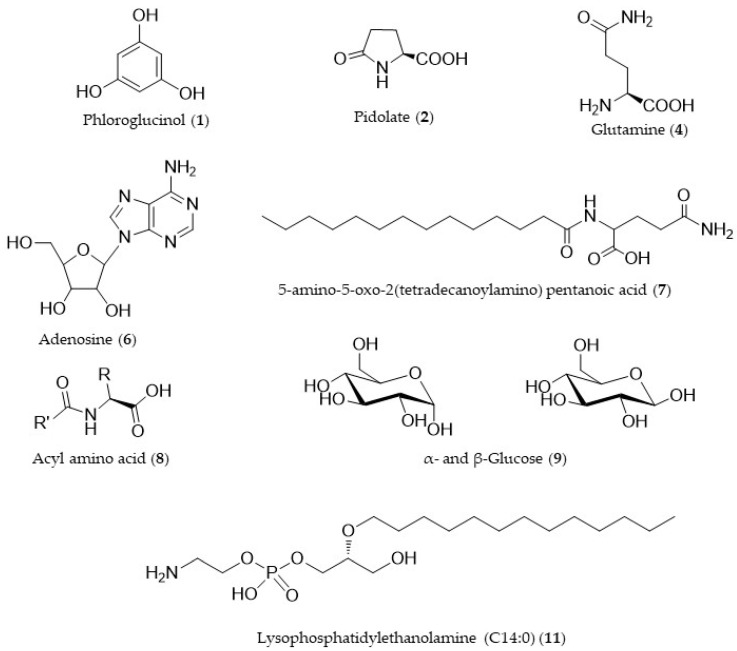
Chemical structures of the compounds identified in group III.

**Figure 7 biology-12-01378-f007:**
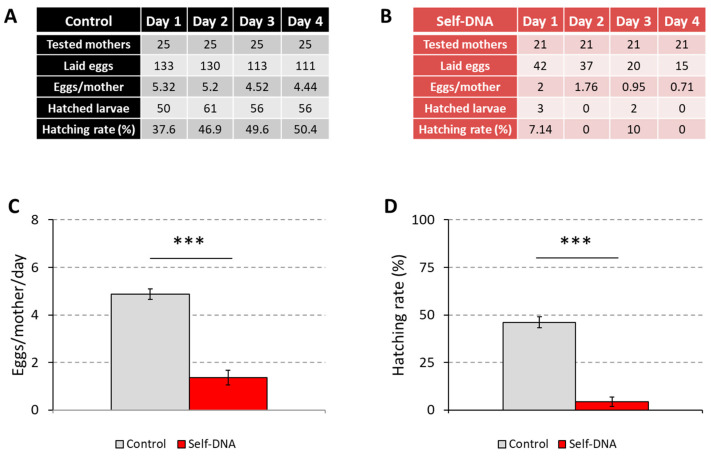
Reproductive traits of *Drosophila melanogaster* females, fed either control (**A**) or a self-DNA (**B**) diet, showing inhibition of egg deposition (**C**), ***: *t*-test for independent samples, *p* < 0.001 and hatching (**D**), ***: *t*-test for independent samples, *p* < 0.001.

**Table 1 biology-12-01378-t001:** List of compounds identified in the analyzed samples.

	Abbreviation	*m* */z*	Rt	Ion	Mol. Formula	Putative Identification	Δppm
**Group I**	His C12:0	338.2437	29.81	M + H^+^	C_18_H_32_O_3_N_3_	Lauryl histidine	−0.350
	His C14:0	366.2753	31.33	M + H^+^	C_20_H_36_O_3_N_3_	Myristoyl histidine	−1.002
	His C14:1	364.2593	31.75	M + H^+^	C_20_H_34_O_3_N_3_	Myristoleyl histidine	−0.462
	His C16:1	392.2908	34.27	M + H^+^	C_22_H_38_O_3_N_3_	Palmitoleyl histidine	0.080
	Tyr C14:1	390.2637	35.26	M + H^+^	C_23_H_36_O_4_N	Myristoleyl tyrosine	−0.474
	Tyr C14:0	392.2797	36.86	M + H^+^	C_23_H_38_O_4_N	Myristoyl tyrosine	−0.978
	Val C14:0	328.2846	38.74	M + H^+^	C_19_H_38_O_3_N	Myristoyl valine	−0.147
	Asp C14:0	344.2420	36.36	M + H^+^	C_18_H_34_O_5_N	Myristoyl aspartic acid	−3.456
	Asp C14:1	342.2274	34.62	M + H^+^	C_18_H_32_O_5_N	Myristoleyl aspartic acid	−0.203
	Phe C14:0	376.2845	39.53	M + H^+^	C_23_H_38_O_3_N	Myristoyl phenylalanine	−0.427
	Glu C14:0	358.2588	36.42	M + H^+^	C_19_H_36_O_5_N	Myristoyl glutamic acid	−0.027
	Glu C14:1	356.2430	34.85	M + H^+^	C_19_H_34_O_5_N	Myristoleyl glutamic acid	−0.420
	Glu C16:0	386.2905	38.82	M + H^+^	C_21_H_40_O_5_N	Palmitoyl glutamic acid	0.777
	Glu C16:1	384.2743	37.15	M + H^+^	C_21_H_38_O_5_N	Palmitoleyl glutamic acid	−0.390
	Glu C18:1	412.3055	39.42	M + H^+^	C_23_H_42_O_5_N	Oleoyl glutamic acid	−0.606
**Group II**	Pro	116.0704	0.77	M + H^+^	C_5_H_10_O_2_N	Proline	−1.681
	Val	118.0861	0.81	M + H^+^	C_5_H_12_O_2_N	Valine	−1.314
	Thr	120.0653	0.90	M + H^+^	C_4_H_10_O_3_N	Threonine	−1.830
	Ile + Leu	132.1017	1.32	M + H^+^	C_6_H_14_O_2_N	Isoleucine/Leucine	−1.554
	Asn	133.0606	0.79	M + H^+^	C_4_H_9_O_3_N_2_	Asparagine	−1.268
	Asp	134.0302	0.83	M + H^+^	C_4_H_8_O_4_N	Aspartic acid	−4.965
	Lys	147.1127	0.67	M + H^+^	C_6_H_15_O_2_N_2_	Lysine	−0.709
	Glu	148.0603	0.79	M + H^+^	C_5_H_10_O_4_N	Glutamic acid	−0.907
	Met	150.0589	0.89	M + H^+^	C_5_H_12_O_2_NS	Methionine	−0.329
	His	156.0768	0.79	M + H^+^	C_6_H_12_O_2_N_3_	Histydine	−0.084
	Phe	166.0862	2.32	M + H^+^	C_9_H_12_O_2_N	Phenylalanine	−0.332
	Arg	175,1188	0.74	M + H^+^	C_6_H_15_O_2_N_4_	Arginine	−0.869
	Tyr	182.0811	1.51	M + H^+^	C_4_H_9_O_4_N_6_	Tyrosine	−0.383
	Trp	205.0942	2.77	M + H^+^	C_11_H_13_O_2_N_2_	Tryptophan	−0.313
**Group III**	1	127.0388	0.71	M + H^+^	C_6_H7O_3_	Phloroglucinol	−1.343
	2	130.0497	0.89	M + H^+^	C_5_H8O_3_N	Pidolate	−1.304
	3	245.6156	37.18	M + 2H^2+^	unidentified	unidentified	
	4	147.0763	0.79	M + H^+^	C_5_H_11_O_3_N_2_	Glutamine	−0.807
	5	191.1248	32.06	M + 2H^2+^	unidentified	unidentified	
	6	268.1040	1.73	M + H^+^	C_10_H_14_O_4_N_5_	Adenosine	−0.113
	7	357.2747	35.84	M + H^+^	C_19_H_37_O_4_N_2_	5-amino-5-oxo-2(tetradecanoylamino) pentanoic acid	−0.235
	8	438.3787	39.21	M + NH_4_^+^	C_21_H_50_O_5_N_4_	Acyl amino acid	2.573
	9	203.0525	0.78	M + Na^+^	C_6_H_12_O_6_Na	Glucose	−0.193
	10	243.0903	30.63	M + H^+^	C_10_H_16_O_3_N_2_P	unidentified	4.920
	11	426.2613	36.42	M + H^+^	C_19_H_41_O_7_NP	Lysophosphatidylethanolamine (C14:0)	−0.506

## Data Availability

The data used to support the findings of this study are included within the article and the Appendix A.

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
