# Peer review of "Self-DNA Inhibition in Drosophila melanogaster Development: Metabolomic Evidence of the Molecular Determinants"

_biology, 2023, doi:10.3390/biology12111378_

Round 1

Reviewer 1 Report

Comments and Suggestions for Authors

Very good research design and thesis skills. Thank you very much for being able to read such an interesting paper. I present only a few minor comments as below.

1. Methods section Line 141: The authors applied 7 mg of control yeast and self DNA library, but the process of determining this amount does not seem clear. It need to explain the minimum amount of self DNA that affects the development of fruit flies and what the maximum amount of self DNA will be.

2. Comments: The results of this study show that the intake of self DNA in  Drosophila slows down a lot of metabolism and reduce fertility. Considering the fields where these results can be applied to human research, I wonder if extracting and ingesting DNA from cancer tissues of cancer patients will reduce the metabolic rate of cancer cells or affect the death of cancer cells. Similarly, it would be nice if you could describe a little more of the expected effects that the findings can be applied to human research.

Reviewer 2 Report

Comments and Suggestions for Authors

The authors applied NMR and LC-MS/MS to analyze the metabolomic evidence of self-DNA inhibition in Drosophila melanogaster development. There are some questions that need to be addressed.

1.       Please show an example of chromatogram from LC-MS analysis in the insects’ culture sample.

2.       Please provide more information about the samples that injected into LC-MS. How were samples prepared or extracted? What solvents were samples in?

3.       In figure 5 please explain what the chemical transformations are between each node.

4.       What is the purpose of displaying group II and III in figure 5 as a molecular network?

Comments on the Quality of English Language

English is good.

Reviewer 3 Report

Comments and Suggestions for Authors

The manuscript “Self-DNA inhibition in Drosophila melanogaster development: metabolomic evidence of the molecular determinants” deserves the highest praise, since the research was done very well and in detail. The only thing that probably needs to be explained in some more detail is the inhibitory effect of Drosophila DNA on its morphogenesis. Only the toxic effect of metabolites is not enough to explain, it seems to me. It is necessary to think about the exact mechanism and describe it at least in the form of an assumption. It is very likely that DNA synthesis is blocked or greatly slowed down during the buildup of DNA metabolites in the Drosophila body. It is likely that DNA polymerase activity is reduced and therefore both fertility and pupation are inhibited in the case of a DNA-rich diet. These are, of course, only probable assumptions, but please consider this mechanism in your manuscript, and, if possible, provide literature data.

All comments made do not diminish the importance of the work done and the manuscript deserves to be published in this journal.
